# Antimalarial Drugs Enhance the Cytotoxicity of 5-Aminolevulinic Acid-Based Photodynamic Therapy against the Mammary Tumor Cells of Mice In Vitro

**DOI:** 10.3390/molecules24213891

**Published:** 2019-10-29

**Authors:** Tomohiro Osaki, Kiwamu Takahashi, Masahiro Ishizuka, Tohru Tanaka, Yoshiharu Okamoto

**Affiliations:** 1Joint Department of Veterinary Medicine, Faculty of Agriculture, Tottori University, Tottori 680-8553, Japan; yokamoto@muses.tottori-u.ac.jp; 2SBI Pharmaceuticals Co., Ltd., Tokyo 106-6020, Japan; kiwtakah@sbigroup.co.jp (K.T.); mishizuk@sbigroup.co.jp (M.I.); tortanaka@sbigroup.co.jp (T.T.)

**Keywords:** 5-aminolevulinic acid, artemisinin, artesunate, antimalarial, artemether, cytotoxicity, photodynamic therapy, reactive oxygen species

## Abstract

Artemisinin and its derivatives, including artesunate (ART) and artemether (ARM), exert anticancer effects in the micromolar range in drug and radiation-resistant cell lines. Artemisinin has been reported to sensitize cervical cancer cells to radiotherapy. In the present study, we determined whether ART and ARM could enhance the cytotoxicity of 5-aminolevulinic acid (5-ALA)-based photodynamic therapy (PDT) against the mammary tumor cells of mice. The corrected PpIX fluorescence intensities in the control, 5-ALA, 5-ALA + ART, and 5-ALA + ARM groups were 3.385 ± 3.730, 165.7 ± 33.45, 139.0 ± 52.77, and 165.4 ± 51.10 a.u., respectively. At light doses of 3 and 5 J/cm^2^, the viability of 5-ALA-PDT-treated cells significantly decreased with ART (*p* < 0.01 and *p* < 0.01) and ARM treatment (*p* < 0.01 and *p* < 0.01). Besides, the number of annexin V-FITC and ethidium homodimer III-positive cells was greater in the 5-ALA-PDT with ARM group than that in the other groups. *N*-acetylcysteine could not significantly inhibit the percentages of apoptotic cells or inviable cells induced by 5-ALA-PDT with ARM. These reactive oxygen species-independent mechanisms might enhance cytotoxicity in 5-ALA-PDT with ARM-treated tumor cells, suggesting that the use of 5-ALA-PDT with ARM could be a new strategy to enhance PDT cytotoxicity against tumor cells. However, as these results are only based on in vitro studies, further in vivo investigations are required.

## 1. Introduction

Breast cancer is the most frequently diagnosed cancer and one of the most fatal diseases affecting women [1]. Surgery, radiation therapy, chemotherapy, and hormone therapy serve as the conventional treatment options for breast cancer. However, photodynamic therapy (PDT), a relatively new approach, has been recently adopted as a treatment strategy. PDT uses the synergistic effects of light and a photosensitizer to kill tumor cells. Photosensitizers, such as porphyrins, chlorins, and phthalocyanines are generally used in clinical PDT [2]. Another example of such a photosensitizer is protoporphyrin IX (PpIX), which is induced by the pro-drug, 5-aminolevulinic acid (5-ALA). Many studies have demonstrated that 5-ALA-induced PpIX can produce reactive oxygen species (ROS) when exposed to light, suggesting that 5-ALA-PDT may be a potential therapeutic modality for the treatment of malignant tumors [3,4]. Although 5-ALA-PDT is effective for malignant tumors, it may be insufficient to cure cancer [5]. Therefore, improving the efficacy of 5-ALA-PDT is needed.

Artemisinin, a chemical compound derived from *Artemisia annua*, has been established as a successful treatment for malaria and is known to exert antineoplastic effects. In both cases, the effects of artemisinin are attributable to the endoperoxide component present in its structure. In the presence of free iron, iron-activated artemisinin induces damage by the releasing of highly alkylating carbon-centered radicals and ROS, such as the superoxide anion and hydroxyl radicals [6,7,8]. Cancer cells are characterized by increased expression of transferrin receptors, which are responsible for the iron uptake and regulation of intracellular concentration. Therefore, iron and heme metabolism might have a relevant role in the selective antitumor activity of artemisinin [9].

Various derivatives of artemisinin, including artesunate (ART), artemether (ARM), dihydroartemisinin (DHA), and arteether, have been identified [10,11]. First-generation semisynthetic artemisinins include the lipophilic artemisinins, arteether, and ARM. ART, however, is a water-soluble derivative [9]. Artemisinin and its derivatives, which are commonly used in malaria therapy, have been reported to exhibit anticancer effects in the micromolar range against drug and radiation-resistant cell lines [12,13,14]. In addition, they exhibit anti-cancer activity against a variety of cancer cells and have been reported to rapidly convert to ROS inside cells, ultimately disrupting cellular functions [15]. ART is well-tolerated by patients, with little toxicity and a lack of evident side effects [16]. Artemisinin has been reported to sensitize cervical cancer cells to radiotherapy, while DHA enhances the effects of 5-ALA-PDT in esophageal cancer cells [17,18]. ART has been reported to kill breast cancer cells via caspase-dependent apoptosis mediated by intracellular ROS accumulation promoted by iron [19]. Previously, we reported that ART displays great potential in enhancing the efficacy of 5-ALA-based sonodynamic therapy for the treatment of cancer [20].

In the present study, we aimed to determine whether ART and ARM could enhance the cytotoxicity of 5-ALA-PDT against an in vitro human cancer model, the mammary tumor cells of mice.

## 2. Results

### 2.1. Protoporphyrin IX Fluorescence in EMT-6 Cells

Microscopically, differences in fluorescence intensity and cell form were not observed among the groups (Figure 1).

The level of 5-ALA-induced PpIX in EMT-6 is shown in Figure 2. After a 4-h incubation with 1 mM 5-ALA, fluorescence intensity was approximately 55-fold higher than that of the control. The corrected PpIX fluorescence intensities of the control, 5-ALA, 5-ALA + ART, and 5-ALA + ARM groups were 3.385 ± 3.730, 165.7 ± 33.45, 139.0 ± 52.77, and 165.4 ± 51.10 a.u., respectively.

### 2.2. Cytotoxicity Analysis of the ART or ARM Dose

Figure 3 shows the viability of 5-ALA-PDT-treated cells in the presence of ART or ARM. At 0 μM of ART or ARM, viability of 5-ALA-PDT-treated cells was dependent on the light dose. In the absence of irradiation (0 J/cm^2^), increasing the concentration of ART or ARM did not significantly affect cell survival. However, at light doses of 3 and 5 J/cm^2^, viability of the 5-ALA-PDT-treated cells was significantly decreased in the presence of ART (*p* < 0.01 and *p* < 0.01) or ARM (*p* < 0.01 and *p* < 0.01), respectively. At a light dose of 3 J/cm^2^, cell viability in 5-ALA-PDT with 7.8 μM ARM group was lower than that in 5-ALA-PDT with 7.8 μM ART group (*p* = 0.0165). At a light dose of 5 J/cm^2^, cell viability in 5-ALA-PDT for 2.0 and 7.8 μM ARM groups was lower than that in 5-ALA-PDT for 2.0 and 7.8 μM ART groups (*p* < 0.0001 and *p* = 0.0002).

### 2.3. Morphological Changes in EMT-6 Cells

Figure 4A–D shows the results of Hoechst 33342 staining of EMT-6 cells incubated with 1 mM 5-ALA for 4 h, irradiated (20 mW/cm^2^, 3 J/cm^2^), and stained with Hoechst 33342 for 6 h after PDT. No signs of apoptosis were observed in the control group. Furthermore, cells treated with 5-ALA-PDT with/without ART displayed slight nuclear condensation, while cells treated with 5-ALA-PDT and ARM displayed more nuclear condensation.

Figure 4E–H shows the results of the morphological evaluation performed with EMT-6. No morphological changes were observed in the control group. However, cells treated with 5-ALA-PDT with/without ART displayed minimal shrinkage and plasma membrane blebbing, while those treated with 5-ALA-PDT and ARM displayed more shrinkage and plasma membrane blebbing.

Figure 5 shows images of EMT-6 cells stained with annexin-V-FITC and ethidium homodimer III and analyzed 6 h after 5-ALA-PDT. Cells treated with 5-ALA-PDT with/without ART were only slightly stained with annexin V-FITC and ethidium homodimer III. In addition, the number of positive cells for annexin V-FITC and ethidium homodimer III staining in the 5-ALA-PDT with ARM group (Figure 5D) was found to be greater than those of the 5-ALA-PDT with/without ART groups (Figure 5B,C). EMT-6 cells treated with 5-ALA-PDT plus ARM were positively stained with annexin V or ethidium homodimer III, indicating late apoptosis or necrosis.

### 2.4. The Effect of an Antioxidant on PDT-Induced Reactive Oxygen Species (ROS) and Apoptosis

To further investigate whether 5-ALA-PDT only or 5-ALA-PDT with ART or ARM induced ROS and apoptosis, EMT-6 cells were treated with 5-ALA-PDT only or 5-ALA-PDT with ART or ARM in the presence (10 mM) or absence of N-acetylcysteine (NAC). As a result, NAC could significantly inhibit ROS induction by 5-ALA-PDT only (*p* = 0.0023) or via 5-ALA-PDT with ARM (*p* < 0.0001) (Figure 6A). In addition, it could significantly inhibit apoptosis’s induction caused by 5-ALA-PDT only (*p* < 0.0001) (Figure 6B). The cytotoxicity due to 5-ALA-PDT only (*p* < 0.0001) or 5-ALA-PDT with ART (*p* = 0.0003) could be significantly inhibited by NAC (Figure 6C). NAC could not significantly inhibit the percentage of apoptotic cells, nor inviable cells induced by 5-ALA-PDT with ARM.

## 3. Discussion

ART has been reported to kill breast cancer cells via caspase-dependent apoptosis mediated by intracellular ROS accumulation promoted by iron [19]. Herein, we hypothesized that ART and ARM could be promising candidates in the search for low-toxicity compounds that could increase the efficacy of 5-ALA-based PDT. In the present study, with a light dose of 3 and 5 J/cm^2^, the viability of 5-ALA-PDT-treated cells significantly decreased following ART (*p* < 0.01 and *p* < 0.01) and ARM (*p* < 0.01 and *p* < 0.01) treatment.

There were no differences in corrected fluorescence intensities and percentages of ROS-positive cells among the 5-ALA-PDT, 5-ALA-PDT with ART, and 5-ALA-PDT with ARM groups. However, the cytotoxicity of 5-ALA-PDT was enhanced when combined with ART or ARM. Besides, the cytotoxicity of 5-ALA with/without ART and ARM-based PDT did not depend on intracellular PpIX. Previously, ART and DHA, as monotherapies, were reported to exert higher anticancer effects against human hepatoma cells than ARM [21]. However, in this study, viability was lower in cells treated with 5-ALA-PDT with ARM than in those treated with 5-ALA-PDT with ART. Hence, the mechanism of ART or ARM cytotoxicity might be complicated when combined with other therapies, such as PDT.

The active fraction of the artemisinin analog is an endoperoxide bridge that generates carbon-centered free radicals or ROS and oxidative stress upon cleavage [22]. ROS plays a significant role in the killing of specific tumor cells via the induction of apoptosis and oxidative DNA damage [23]. The intracellular bioactivation of the endoperoxide to carbon-centered radicals was found to be dependent on cellular heme with the addition of PpIX, a precursor of heme that could increase heme synthesis to levels approximately 2.5-fold greater than normal, thereby significantly increasing the sensitivity to ART-induced cytotoxicity [24]. ART treatment alone for 24 h resulted in an IC_50_ of approximately 2 μM for human colorectal cancer cells, while the addition of 5-ALA lowered the IC_50_ by approximately 10-fold to around 200 nM [25]. As addition of 5-ALA increased intracellular heme level and enhanced ART activation in cancer cells, ART/5-ALA combination therapy could be more effective than ART monotherapy [25]. In this study, 5-ALA-PDT with ART or ARM was more effective than 5-ALA-PDT alone at 3 and 5 J/cm^2^. The cytotoxicity of 5-ALA-PDT with ART or ARM might be endoperoxide-dependent. However, further studies in mouse models of EMT6 mammary tumors are required to evaluate the in vivo photodynamic effects of 5-ALA-PDT-ARM.

Although NAC significantly inhibited the generation of ROS induced by 5-ALA-PDT with/without ARM, NAC could not significantly inhibit the percentage of apoptotic cells and inviable cells induced by 5-ALA-PDT with ARM. Hence, we speculate that mechanisms other than the generation of ROS are involved in cell death induced by 5-ALA-PDT with ARM. A recent report revealed that the major metabolites of artemisinin and its derivative, DHA, induced the binding of the nuclear protein, murine double minutes 2, and downregulated the levels of the cell surface transferrin receptor to inhibit p53 ubiquitination [9,21,23,26]. Hence, such ROS-independent mechanisms might enhance the efficacy of 5-ALA-PDT with ARM.

## 4. Materials and Methods

### 4.1. Cell Line and Culture Conditions

Mouse mammary tumor EMT6 cells were supplied by Professor Yoshihiro Uto of Tokushima University (Tokushima, Japan). The cells were maintained as an adherent monolayer culture, and incubated in 5% CO_2_ at 37 °C. RPMI 1640 medium (Invitrogen, Carlsbad, CA, USA) supplemented with 10% heat-inactivated fetal bovine serum (Nichirei Biosciences Inc., Tokyo, Japan) and PSN (5 mg/mL penicillin, 5 mg/mL streptomycin, and 10 mg/mL neomycin; Invitrogen) was employed as the culture medium. For the experiments, cells were harvested from near-confluent cultures via brief exposure to a solution containing 0.25% trypsin and 1 mmol/L EDTA-4Na with phenol red (Invitrogen). Trypsinization was terminated using RPMI 1640 medium containing 10% fetal bovine serum. Thereafter, cells were centrifuged and re-suspended in RPMI 1640 medium. Trypan blue staining was used to assess cell viability.

### 4.2. Chemicals

5-ALA was donated by SBI Pharma (Tokyo, Japan). A stock solution of 100 mM 5-ALA in phosphate-buffered saline (PBS), stored at 4 °C, was used for the in vitro experiments. Stock solutions of ART and ARM (Tokyo Chemical Industry Co., Ltd., Tokyo, Japan) at 1 mM were prepared with PBS and 1% (*v*/*v*) DMSO and diluted to the final concentrations indicated in the cell cultures.

### 4.3. Protoporphyrin IX Fluorescence in EMT-6 Cells

To investigate the formation of PpIX from 5-ALA, PpIX fluorescence was examined using an Olympus Fluoview FV1000 (Olympus Co., Tokyo, Japan) and read with a fluorescence spectrometer SH-9000Lab (Hitachi High-Technologies Co., Tokyo, Japan).

First, 1 × 10^5^ EMT-6 cells were seeded in 35-mm Petri dishes containing 2 mL of cultivation medium. After 24 h of incubation, the dishes were divided into the following four groups: (1) Control group (no treatment), (2) 5-ALA group (treated with 1 mM 5-ALA), (3) 5-ALA + ART group (treated with 1 mM 5-ALA and 7.8 μM ART), and (4) 5-ALA + ARM group (treated with 1 mM 5-ALA and 7.8 μM ARM). Cells were incubated with 1 mM 5-ALA and concentrations of 7.8 μM ART or ARM for 4 h.

After washing with PBS, PpIX fluorescence was examined using an Olympus Fluoview FV1000. To obtain the cell extracts, cells were harvested by trypsinization, and centrifuged at 300× *g* for 5 min at room temperature. Whole-cell extracts were prepared with 50 μL Triton buffer (1% Triton X-100 in PBS). The fluorescence of cell lysates was read at Ex = 405 nm and Em = 635 nm with a fluorescence spectrometer SH-9000Lab.

### 4.4. Cytotoxicity Analysis of the ART or ARM Dose

A total of 4–5 × 10^4^ of primary cells was seeded into each well of 96-well plates (Corning Inc., Corning, NY, USA) prior to overnight incubation. Cells were then incubated with 1 mM 5-ALA and various concentrations of ART or ARM (0, 2.0, 7.8, 31.3, 125, and 500 μM) for 4 h. After washing with fresh medium, cells were irradiated with a 630-nm light (20 mW/cm^2^, 10 J/cm^2^) emitted by LED lights (Pleiades Technologies LLC, Fukuoka, Japan). Subsequently, the cells were re-incubated for 24 h in the dark. Cell viability was examined by using Cell Counting Kit-8 in accordance with the manufacturer’s instructions.

### 4.5. Morphological Changes in EMT-6 Cells

A total of 1 × 10^5^ EMT-6 cells were seeded in 35-mm Petri dishes containing 2 mL of cultivation medium. After 24 h of incubation, the dishes were divided into the following four groups: (1) Control group (no treatment), (2) 5-ALA-based PDT group (treatment with 1 mM 5-ALA and irradiated with a light dose of 3 J/cm^2^), (3) 5-ALA + ART-based PDT group (treatment with 1 mM 5-ALA and 7.8 μM ART, and irradiated with a light dose of 3 J/cm^2^), and (4) 5-ALA + ARM-based PDT group (treatment with 1 mM 5-ALA and 7.8 μM ARM, and irradiated with a light dose of 3 J/cm^2^).

Cells were incubated with 1 mM 5-ALA and 7.8 μM ART or ARM for 4 h. After washing with fresh media, cells were irradiated with a 630-nm laser light (20 mW/cm^2^, 3 J/cm^2^) emitted by LED lights. After 6 h of PDT, cells were stained using the Promokine Apoptotic/Necrotic/Healthy cell detection kit, according to the manufacturer’s instructions. Subsequently, cells were stained with Hoechst 33342. Nuclear morphology was examined using an Olympus BX51 (Olympus Co., Tokyo, Japan). To assess apoptosis and necrosis, cells were stained with annexin V-fluorescein isothiocyanate (FITC) and ethidium homodimer III at 6 h after laser irradiation. Annexin V was used to detect phosphatidylserine on the external membrane of apoptotic cells. Cells were analyzed by Olympus Fluoview FV1000 using the FITC and Texas Red filter settings.

### 4.6. Analysis of Reactive Oxygen Species (ROS) and Apoptosis Induced by PDT

EMT-6 cells were seeded at 1 × 10^4^ cells in 35‑mm petri dishes containing 2 mL of culture medium. Following 24 h of incubation, cells were divided into the following four groups: (1) Control group (no treatment), (2) 5-ALA-based PDT group (treatment with 1 mM 5-ALA and irradiated with a light dose of 3 J/cm^2^), (3) 5-ALA + ART-based PDT group (treatment with 1 mM 5-ALA and 7.8 μM ART, and irradiated with a light dose of 3 J/cm^2^), and (4) 5-ALA + ARM-based PDT group (treatment with 1 mM 5-ALA and 7.8 μM ARM, and irradiated with a light dose of 3 J/cm^2^).

Cells were incubated with 1 mM 5-ALA and 7.8 μM ART or ARM for 4 h in the presence (10 mM) or absence of the antioxidant, *N*-acetylcysteine (NAC). After washing with fresh media, cells were irradiated with a 630-nm laser light (20 mW/cm^2^, 3 J/cm^2^) emitted by LED lights.

To determine the percentage of cells that were negative (healthy cells) and positive for ROS (cells containing ROS), ROS generation was assessed 6 h after laser irradiation using the Muse Oxidative Stress kit (EMD Millipore Co., Billerica, MA, USA) according to the manufacturer’s protocols.

The total apoptotic cells of early and late stages and cell viability were assessed 6 h after LED irradiation using the Muse Annexin V (EMD Millipore Co.) and Dead Cell Assay kit (EMD Millipore Co.), according to the manufacturers’ protocols.

Cells were harvested for the experiments and single-cell suspensions were loaded onto the Muse Cell Analyzer (EMD Millipore Co.).

### 4.7. Statistical Analysis

Data were analyzed by Dunnett and Sidak’s multiple comparisons test. *p* < 0.05 was considered to indicate a statistical significance. Statistical analyses were performed using GraphPad Prism software (version 6.0; GraphPad Software, Inc., La Jolla, CA, USA).

## 5. Conclusions

Herein, we identified that ARM could enhance 5-ALA-PDT-induced EMT-6 cell death. To our knowledge, this is the first study to evaluate 5-ALA-PDT with ARM as a new therapeutic strategy to enhance the cytotoxicity of PDT against tumor cells. However, because the results are only based on in vitro studies, in vivo studies should be performed. In addition, further studies are needed to investigate the unknown mechanisms of action in vitro and the difference in antitumor efficacy against different tumor cell lines.

## Figures and Tables

**Figure 1 molecules-24-03891-f001:**
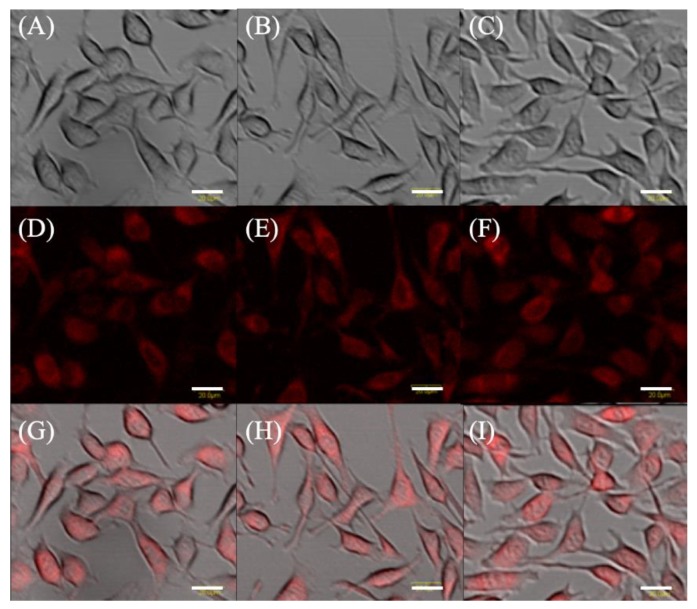
Protoporphyrin IX fluorescence in EMT-6. Cells were incubated with 1 mM 5-aminolevulinic acid (5-ALA) (**A**,**D**,**G**), and 1 mM 5-ALA plus concentrations of 7.8 μM artesunate (**B**,**E**,**H**) or artemether (**C**,**F**,**I**) for 4 h. Transmission, fluorescence, and merged images of EMT-6. Upper panel (**A**–**C**): Transmitted light images. Middle panel (**D**–**F**): Fluorescence images. Lower panel (**G**–**I**): Merged images. Scale bar = 20 μm.

**Figure 2 molecules-24-03891-f002:**
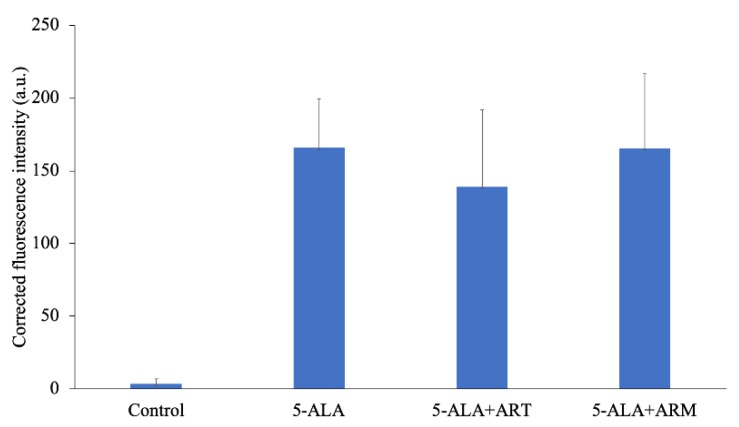
Corrected protoporphyrin IX fluorescence intensity in EMT-6 cells. Cells were incubated with 1 mM 5-aminolevulinic acid (5-ALA), 1 mM 5-ALA, and concentrations of 7.8 μM artesunate (ART) or artemether (ARM) for 4 h. Results are presented as means ± standard deviations (*n* = 5).

**Figure 3 molecules-24-03891-f003:**
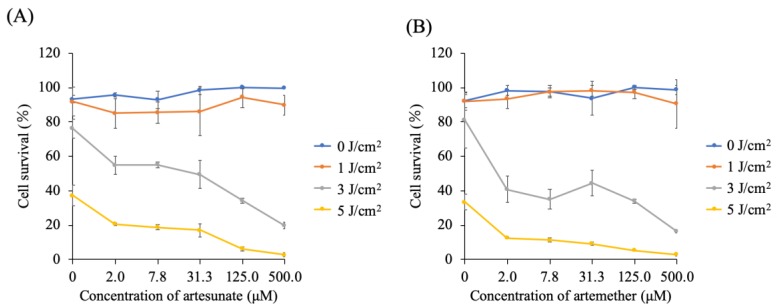
Cytotoxicity analysis of the artesunate (ART) or artemether (ARM) dose. Cells were incubated for 4 h with 1 mM 5-aminolevulinic acid (5-ALA) and various concentrations (0, 2, 7.8, 31.3, 125, and 500 μM) of (**A**) ART or (**B**) ARM. After washing with fresh medium, cells were irradiated with a 630-nm light (20 mW/cm^2^, 10 J/cm^2^) emitted by LED lights. At fluorescence intensities of 3 and 5 J/cm^2^, the viability of the 5-ALA-photodynamic therapy (PDT)-treated cells was significantly decreased in the presence of ART (*p* < 0.01 and *p* < 0.01) or ARM (*p* < 0.01 and *p* < 0.01). Results are presented as means ± standard deviations (*n* = 5).

**Figure 4 molecules-24-03891-f004:**
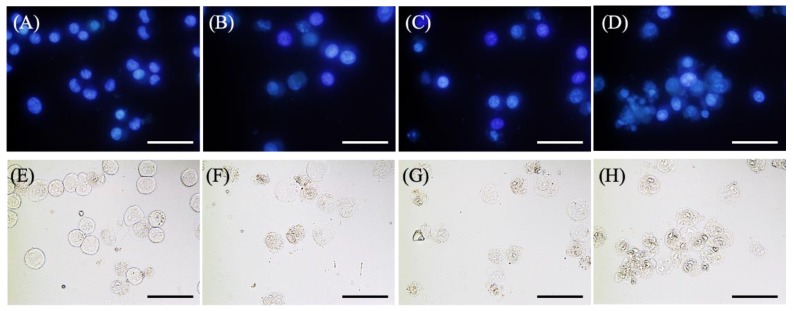
Morphological changes in EMT-6 cells 6 h after PDT with artesunate (ART) or artemether (ARM). EMT-6 cells were incubated with 1 mM 5-aminolevulinic acid (5-ALA) for 4 h prior to washing and exposure to a 635-nm LED light. Subsequently, cells were incubated for an additional 6 h before Hoechst 33342 staining. The 5-ALA-PDT with ARM group had a marked increase in condensation of nuclear chromatin relative to the other groups. Images represent the (**A**,**E**) control, (**B**,**F)** 5-ALA-PDT, (**C**,**G**) 5-ALA-PDT and ART, and (**D**,**H**) 5-ALA-PDT and ARM groups. Upper panel (**A**–**D**): Fluorescence images. Lower panel (**E**–**H**): Transmitted light images. Scale bar = 50 μm.

**Figure 5 molecules-24-03891-f005:**
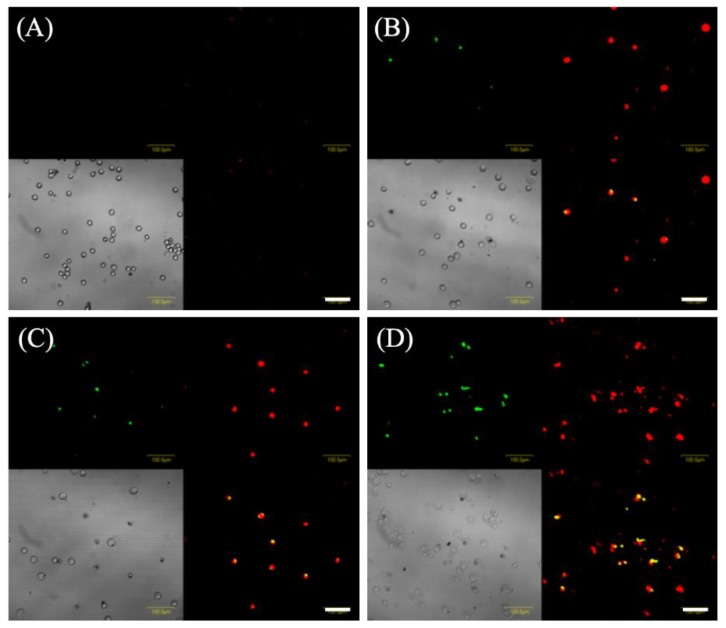
Fluorescent staining to detect cell death in EMT-6 cells. EMT-6 cells were incubated with 1 mM 5-aminolevulinic acid (5-ALA) for 4 h prior to washing and exposure to a 635-nm LED light. Cells were further incubated for 6 h before staining with annexin V (green) and ethidium homodimer III (red). Images represent the (**A**) control, (**B**) 5-ALA-PDT, (**C**) 5-ALA-PDT and artesunate, and (**D**) 5-ALA-PDT and artemether groups. Transmitted light images. Scale bar = 100 μm.

**Figure 6 molecules-24-03891-f006:**
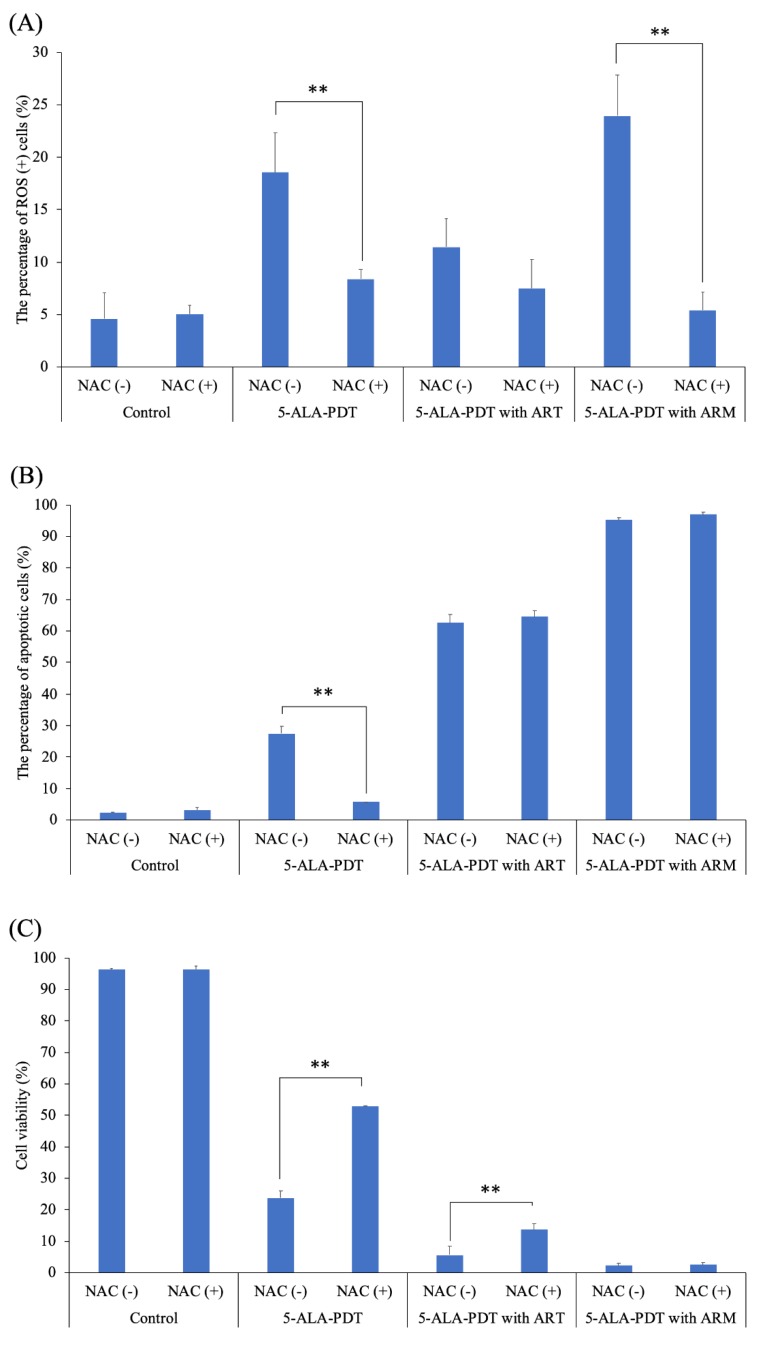
The percentage of ROS-positive cells in the presence or absence of N-acetylcysteine (NAC). EMT-6 cells were treated with 5-aminolevulinic acid (5-ALA)-PDT only or 5-ALA-PDT with artesunate (ART) or artemether (ARM) in the presence (10 mM) or absence of NAC. ROS’ induction by 5-ALA-PDT only (*p* = 0.0023) or 5-ALA-PDT with ARM (*p* < 0.0001) was significantly inhibited by NAC (**A**). Apoptosis’s induction by 5-ALA-PDT only (*p* < 0.0001) was significantly inhibited by NAC (**B**). Cytotoxicity, exhibited by 5-ALA-PDT only (*p* < 0.0001) or 5-ALA-PDT with ART (*p* = 0.0003), was significantly inhibited by NAC (**C**). Data were analyzed using Sidak’s multiple comparison test (** *p* < 0.01; NAC (−) vs. NAC (+)). Results are presented as means ± standard deviations.

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
