# Peer review of "Antimalarial Drugs Enhance the Cytotoxicity of 5-Aminolevulinic Acid-Based Photodynamic Therapy against the Mammary Tumor Cells of Mice In Vitro"

_molecules, 2019, doi:10.3390/molecules24213891_

Round 1

Reviewer 1 Report

The paper entitled „Antimalarial drugs enhance cytotoxicity of 5-aminolevulinic acid-based photodynamic therapy against mouse mammary tumor cells in vitro” by T. Osaki, K. Takahashi, M. Ishizuka, T. Tanaka and Y. Okamoto reports potentiation of PP-IX (derivied from pro-drug 5-ALA) activity against cancer by antimalarial drugs – artesunate and artemether. Authors have studied formation PP-IX from pro-drug 5-ALA; cytotoxicity in the present of antimalarial drugs; cancer cells morphological changes after treatment. They have noticed enhanced activity of PP-IX. In my opinion this paper is interesting and can be published after minor revision.

We could ask authors for in vivo trials, but it is not necessary. On the other hand, is it good from ethical point of view test therapies, which are not studied well in in vitro model? We need more studies in in vitro model i.e. against another cancer cell lines.

line 154, "....at the fluorescence of 3 and 5 J/cm2, the viability ...." I think that instead term "fluorescence" should be "light dose"

Author Response

Reviewer 1

We could ask authors for in vivo trials, but it is not necessary. On the other hand, is it good from ethical point of view test therapies, which are not studied well in in vitro model? We need more studies in in vitro model i.e. against another cancer cell lines.

Response: We appreciate this thoughtful suggestion. In the present study, we assessed the efficacy of antimalarial drugs, artesunate and artemether, for the 5-ALA-based PDT. We identified that artemether could enhance 5-ALA-based PDT-induced EMT-6 cell death. In future, we would like to investigate the difference in antitumor efficacy against different tumor cell lines. We have added “the difference in antitumor efficacy against different tumor cell lines” to the conclusion.

line 154, "....at the fluorescence of 3 and 5 J/cm2, the viability ...." I think that instead term "fluorescence" should be "light dose"

Response: We appreciate this advice and have changed “fluorescence” to “light dose.”

Reviewer 2 Report

The present study demonstrates the potential efficiency of Artemisinin derivatives (artesunate and artemether) in combination with 5-ALA PDT against mouse mammary monolayer cells. The authors provided all needed controls and showed how the cytotoxicity depends on the fluence rate and concentration of Artemisinin derivatives. The mechanisms of cell death were also addressed. The MS is clearly written providing up to date literature references.

However, the authors should answer several principal questions before their paper could be considered for publication :

The apoptosis assay is confusing. According to the manufacturer protocols, Annexin-V stains both apoptotic and necrotic cells, while ethidium homodimer III stains only necrotic ones. In the Figure 5, there are the cells which are stained for ethidium homodimer III only; this is confusing since if membrane is disrupted Annexin-V should stain the Phosphatidylserine on the inner part of plasma membrane.

Figure 6B represents more than 90% of apoptotic cells in the samples with 5-ALA PDT + ARM. Could the authors confirm that only apoptotic cells were considered here but not apoptotic and necrotic cells?

There is no correlation between the data presented in Figure 3 and Figure 6. According to the Figure 3, cell viability 24 hours after PDT of samples treated with 7.8 mM ART and ARM at 5-ALA PDT at 3 J cm-2 is about 50% and 40% respectively. The 5-ALA PDT at 3 J cm-2 control samples demonstrated 80% viable cells.

On the other hand, the cell viability 6 hours after PDT (at 3 J cm-2), presented in Figure 6C is 25%, 10% and 5% for 5-ALA PDT only, PDT + ART and PDT+ ARM respectively. Please clarify this point.

Minor suggestions:

The addition to the discussion of chemical structures of Artemisinin derivatives could simplify the explanation of their action mechanism.

Figures 4 and 5 could be improved by the addition of the magnification of certain cells corresponding to the discussed details.

Author Response

Reviewer 2 

The apoptosis assay is confusing. According to the manufacturer protocols, Annexin-V stains both apoptotic and necrotic cells, while ethidium homodimer III stains only necrotic ones. In the Figure 5, there are the cells which are stained for ethidium homodimer III only; this is confusing since if membrane is disrupted Annexin-V should stain the Phosphatidylserine on the inner part of plasma membrane.

Figure 6B represents more than 90% of apoptotic cells in the samples with 5-ALA PDT + ARM. Could the authors confirm that only apoptotic cells were considered here but not apoptotic and necrotic cells?

Response: We appreciate this thoughtful suggestion. We have re-written the sentence to say “ The total apoptotic cells of early and late stages and cell viability were assessed...”

There is no correlation between the data presented in Figure 3 and Figure 6. According to the Figure 3, cell viability 24 hours after PDT of samples treated with 7.8 mM ART and ARM at 5-ALA PDT at 3 J cm-2 is about 50% and 40% respectively. The 5-ALA PDT at 3 J cm-2 control samples demonstrated 80% viable cells. On the other hand, the cell viability 6 hours after PDT (at 3 J cm-2), presented in Figure 6C is 25%, 10% and 5% for 5-ALA PDT only, PDT + ART and PDT+ ARM respectively. Please clarify this point.

Response: We appreciate this thoughtful suggestion. The difference in cell viability might be related to the different method and the timing of the cell viability assay.

In Figure 3, Cell Counting Kit-8 was used for the cell viability assay. Cell Counting Kit-8 measures the metabolic activity of living cells; hence, the data do not verify cell death. In contrast, in Figure 6, the Muse Annexin V and Dead Cell Assay kit was used for the cell viability assay. This assay evaluated structural integrity of the cell membrane. Therefore, these data cannot be directly compared.

Moreover, in Figure 3, the assay was done 24 hours after PDT, while in Figure 6, the assay was done 6 hours after PDT. Tumor cells likely regrew between these assays.

Minor suggestions:

The addition to the discussion of chemical structures of Artemisinin derivatives could simplify the explanation of their action mechanism.

Response: We appreciate this suggestion. We have re-written the section as follows:

“In both cases, the effects of artemisinin are attributable to the endoperoxide component in its structure. In the presence of free iron, iron-activated artemisinin induces damage by releasing highly alkylating carbon-centered radicals and ROS such as the superoxide anion and hydroxyl radicals [6-8]. Cancer cells are characterized by increased expression of transferrin receptors, which are responsible for iron uptake and regulation of intracellular concentration. Therefore, iron and heme metabolism might have a relevant role in the selective antitumor activity of artemisinin [9].”

Figures 4 and 5 could be improved by the addition of the magnification of certain cells corresponding to the discussed details.

Response: In previous peer review, we were recommended to show cell images at this magnification level. Therefore, we unfortunately do not have cell images at a higher magnification than this. However, complete tumor cells are shown.

Round 2

Reviewer 2 Report

I suggest to accept the manuscript in present form